# Risk Factors, Pathogens, and Outcomes of Ventilator-Associated Pneumonia in Non-Cardiac Surgical Patients: A Retrospective Analysis

**DOI:** 10.3390/microorganisms12071422

**Published:** 2024-07-13

**Authors:** Po-Hsun Chang, Ting-Lung Lin, Ying-Ju Chen, Wei-Hung Lai, I-Ling Chen, Hui-Chuan Chang, Yu-Cheng Lin, Yu-Hung Lin, Wei-Feng Li, Yueh-Wei Liu, Chih-Chi Wang, Shih-Feng Liu

**Affiliations:** 1Department of Pharmacy, Kaohsiung Chang Gung Memorial Hospital, Kaohsiung 833, Taiwan; pohsunchang@cgmh.org.tw (P.-H.C.); iling10@cgmh.org.tw (I.-L.C.); 2Chang Gung University College of Medicine, Kaohsiung 833, Taiwan; tinolin@adm.cgmh.org.tw (T.-L.L.); miraclesmoke@cgmh.org.tw (Y.-J.C.); zetaforce@hotmail.com (W.-H.L.); elaine11142@cgmh.org.tw (H.-C.C.); linyucheng@cgmh.org.tw (Y.-C.L.); adrianlin107@cgmh.org.tw (Y.-H.L.); webphone@cgmh.org.tw (W.-F.L.); anthony0612@cgmh.org.tw (Y.-W.L.); ufel4996@ms26.hinet.net (C.-C.W.); 3Department of Surgery, Kaohsiung Chang Gung Memorial Hospital, Kaohsiung 833, Taiwan; 4School of Pharmacy, Kaohsiung Medical University, Kaohsiung 807, Taiwan; 5Department of Respiratory Therapy, Kaohsiung Chang Gung Memorial Hospital, Kaohsiung 833, Taiwan; 6Division of Pulmonary and Critical Care Medicine, Department of Internal Medicine, Kaohsiung Chang Gung Memorial Hospital, Kaohsiung 833, Taiwan

**Keywords:** ventilator-associated pneumonia, healthcare-associated pneumonia, non-cardiac surgery, surgical critical illness, surgical intensive care unit

## Abstract

Ventilator-associated pneumonia (VAP) is a critical hospital-acquired infection following non-cardiac surgeries, leading to poor outcomes. This study identifies VAP risk factors in non-cardiac surgical patients and determines the causative pathogens. A retrospective analysis with 1:4 propensity-score matching was conducted on patients in a surgical intensive care unit (ICU) from 2010 to 2020 at a private tertiary medical center. Among 99 VAP patients, the mortality rate was 64.7%. VAP risk factors included prolonged mechanical ventilation (odds ratio [OR] 6.435; *p* < 0.001), repeat intubation (OR 6.438; *p* < 0.001), lower oxygenation levels upon ICU admission (OR 0.950; *p* < 0.001), and undergoing gastrointestinal surgery (OR 2.257; *p* = 0.021). The 30-day mortality risk factors in the VAP group were late-onset VAP (OR 3.450; *p* = 0.022), inappropriate antibiotic treatment (OR 4.083; *p* = 0.041), and undergoing gastrointestinal surgeries (OR 4.776; *p* = 0.019). Nearly half of the Gram-negative infections were resistant strains, and a third were polymicrobial infections. Non-cardiac surgical patients with VAP face adverse hospital outcomes. Identifying high-risk patients and understanding VAP’s resistant and microbial nature are crucial for appropriate treatment and improved health outcomes.

## 1. Introduction

Around 310 million major surgeries are performed yearly worldwide [1]. Despite notable progress in infection management and enhancements in postoperative care, bacterial infections pose a significant challenge following surgical procedures. These infections contribute to patient distress, escalated mortality rates, and increased hospital expenditures [2,3]. Ventilator-associated pneumonia (VAP), categorized as a healthcare-associated infection, leads to adverse hospital experiences and bears a substantial economic burden [4,5].

VAP is particularly concerning due to its association with prolonged hospital stays, increased use of healthcare resources, and higher costs of care. It is known to develop in patients who require mechanical ventilation (MV) for more than 48 h, making it a significant issue in intensive care units (ICUs). Despite extensive research on VAP in cardiac surgical patients, which has shown poor outcomes and high mortality rates [6,7], there are limited comprehensive data on VAP in non-cardiac surgical patients. This lack of data creates a gap in our understanding of the full impact of VAP on a broader surgical population.

Non-cardiac surgical patients represent a diverse group with varying surgical complexities and comorbidities, which may influence the incidence and outcomes of VAP differently compared to cardiac surgical patients. The complexity and variety of non-cardiac surgeries, including abdominal, orthopedic, and neurosurgical procedures, present unique risks and challenges in the management of postoperative infections. Understanding these specific risk factors is crucial for developing targeted preventive and therapeutic strategies.

This study posits that non-cardiac surgical patients afflicted with VAP also encounter unfavorable clinical outcomes in comparison to their counterparts without VAP. Its primary objective is to explore the in-hospital mortality, risk factors associated with VAP in the non-cardiac surgical patient cohort, and predictors of mortality after acquiring VAP. Furthermore, the study seeks to delineate the bacterial pathogens implicated in these instances of VAP. By identifying these factors, the study aims to contribute to improved clinical management and outcomes for non-cardiac surgical patients at risk of developing VAP.

## 2. Methods

### 2.1. Study Design, Setting, and Participants

This retrospective study examined adult patients (aged 20 and above) who underwent surgery at Kaohsiung Chang Gung Memorial Hospital, Taiwan, and were admitted to the surgical intensive care unit (SICU) from January 2010 to December 2020. Exclusions comprised patients with ICU stays shorter than three days, those on MV for less than 48 h, or those undergoing heart surgery. Patients were divided into VAP and non-VAP groups. Only the initial VAP event per hospital admission was analyzed for patients with multiple occurrences. A propensity-match method was used to establish a comparison cohort, balancing covariates between VAP and non-VAP patients to facilitate intergroup differences analysis [8,9]. In the non-VAP group, the ‘index event’ was the first day of MV in the ICU. The diagnosis of VAP was determined by the research team based on clinical criteria and not solely reliant on ICD codes.

### 2.2. Data Collection

The Chang Gung Research Database, one of Taiwan’s most comprehensive multi-institutional medical and healthcare data collections [10], served as the primary data source for this study. This database offers a deidentified dataset, making it a valuable resource for clinical research due to its extensive volume of patient information. This research thoroughly collected patient data, including demographic information (age, gender, and other vital demographic details), Charlson’s Comorbidity Index [11], Sequential Organ Failure Assessment (SOFA) score [12], ICU interventions (MV, tracheostomy, repeat intubation, total parenteral nutrition (TPN), hemodialysis, extracorporeal membrane oxygenation (ECMO), blood transfusion, and catheter insertion), pre-VAP medications (antibiotics, chemotherapy, immunosuppressants, and steroids), American Society of Anesthesiologists (ASA) score [13], surgical details (surgical wound classification [14], operative time, number of operations, and surgical sites), laboratory parameters on ICU admission (PaO_2_/FiO_2_, serum albumin, alanine aminotransferase, creatinine, hemoglobin, platelet count, and total bilirubin), and outcomes of interest (duration of mechanical ventilation, length of ICU and hospital stays, and 30-day and in-hospital mortality). The study period for patient admission was from January 2010 to December 2020.

### 2.3. Definitions

Healthcare-Associated Pneumonia (HAP): Diagnosed in patients who have been hospitalized for more than 48 h and meet all the following criteria [15]:

Chest Imaging Findings: Presence of infiltration, consolidation, or cavitation indicated in chest imaging.

Symptomatic Criteria: The patient exhibits at least one of the following symptoms: a fever exceeding 38 °C, leukopenia (white blood cell count ≤ 4000 WBC/mm^3^), or leukocytosis (white blood cell count ≥ 12,000 WBC/mm^3^).

Positive Respiratory Sample: A positive result in at least one respiratory sample, such as sputum collection or bronchial lavage.

Impaired Gas Exchange: Worsening of gas exchange demonstrated by oxygen saturation (SatO_2_) below 90% or a partial pressure of oxygen to fraction of inspired oxygen ratio (PaO_2_/FiO_2_) of 240 or less.

Ventilator-Associated Pneumonia (VAP): Diagnosed in patients who met the criteria for HAP and had additional specific conditions related to MV:

Duration of MV: The patient must have been on MV for more than two consecutive days.

Timing of MV: The mechanical ventilator must have been in place either on the day of the event or the day before [15].

Gastrointestinal Surgery: Refers to abdominal surgery involving the gastrointestinal tract, liver, biliary tract, or pancreas.

Late-Onset VAP: Defined as VAP onset five days after MV [16].

Resistant Bacterial Infection: Non-susceptible to at least one agent in at least one antimicrobial category. Non-susceptibility refers to resistant, intermediate, or non-susceptible in vitro antimicrobial susceptibility results [17].

Polymicrobial Bacterial Infection: Acute diseases caused by various combinations of bacteria [18].

Inappropriate Antibiotic Therapy: Empiric antibiotic therapy was deemed inappropriate if it failed to include at least one antibiotic proven effective in vitro against the identified pathogens within three days after an index VAP event.

Critical Condition: Unstable vital signs noted during an operation.

Ethics: The study received approval from the Institutional Review Board of Kaohsiung Chang Gung Memorial Hospital, Taiwan (202200091B0C601). Due to the anonymous nature of the data analysis, the requirement for informed consent was waived.

### 2.4. Statistical Analysis

Continuous variables were represented as means ± standard deviations, while categorical variables were shown as numbers and percentages. The Student’s *t*-test was used for the analysis of continuous variables. For categorical variables, either the chi-square test or Fisher’s exact test was employed, depending on the data. A 1:4 propensity-score matching was conducted, factoring in age, sex, body mass index, and Charlson score. Propensity-score matching was done using a logistic regression model to estimate the propensity scores, and the matching was performed using the nearest-neighbor method without replacement. This method was used to balance significant variables between the VAP and non-VAP groups. A *p* value of <0.05 was considered significant for including variables in the multivariable logistic regression. A logistic regression model with a stepwise procedure was utilized to identify risk factors associated with acquiring VAP and mortality following VAP. The distribution of pathogens was analyzed, focusing on different infection sites within the VAP group. For analyzing 30-day survival, the Kaplan–Meier method was applied. The log-rank test was used to generate the *p* value. All tests were two-sided, with *p* < 0.05 as the statistical significance threshold. Analyses were conducted using SAS EG version 5 (SAS Institute, Cary, NC, USA).

## 3. Results

Over eleven years, 8615 surgical patients were admitted to the SICU at Kaohsiung Chang Gung Memorial Hospital, of which 1763 non-cardiac surgical patients who had an ICU stay exceeding three days and required MV for more than 48 h were identified. Of these, 99 patients (5.6%) were classified into the VAP group. The remaining 1583 patients were categorized as the non-VAP group, as illustrated in Figure 1.

### 3.1. Propensity Matched

Following a 1:4 propensity-score matching process, 396 patients were selected for the non-VAP group. Compared with the matched non-VAP group, the VAP group had a higher SOFA score upon ICU admission (6.8 ± 3.5 vs. 6.0 ± 3.6, *p* = 0.039), a higher incidence of requiring MV for over seven days (61.6% vs. 28.3%, *p* < 0.001), repeat intubation (28.3% vs. 17.2%, *p* = 0.012), requiring TPN support (43.4% vs. 30.3%, *p* < 0.013), having a critical condition during operation (59.6% vs. 46.2%, *p* = 0.017), and needing gastrointestinal surgery (77.8% vs. 67.4%, *p* < 0.045), and a lower PaO_2_/FiO_2_ ratio upon ICU admission (280.3 ± 116.6 mmHg vs. 348.6 ± 153.6 mmHg, *p* < 0.001) (Table 1).

### 3.2. Multivariable Analysis for Acquiring VAP

In multivariable analysis, the need for MV for more than 7 days (odds ratio [AOR] 6.435, 95% confidence interval [CI] 3.15–13.146; *p* < 0.001), repeat intubation within 14 days (AOR 6.438, 95% CI 2.934–14.127; *p* < 0.001), undergoing gastrointestinal surgery (AOR 2.257, 95% CI 1.13–4.507, *p* = 0.021), and a lower PaO_2_/FiO_2_ ratio upon ICU admission (AOR 0.950, 95% CI 0.992–0.998; *p* < 0.001) were independent risk factors of acquiring VAP after non-cardiac surgery (Table 2).

### 3.3. Distribution of Bacterial Pathogens According to Infection Sites in the VAP Group

In the VAP group, patients presented with various bacterial infections at different anatomical sites. Gram-negative bacteria were most commonly found in pulmonary isolations (73%), followed by the abdomen (49%), wounds (46%), and blood (38%). The most frequent Gram-negative bacteria in pulmonary isolations included *Pseudomonas aeruginosa*, carbapenem-resistant *Pseudomonas aeruginosa*, *Acinetobacter baumannii*, carbapenem-resistant *Acinetobacter baumannii*, and *Stenotrophomonas maltophilia*. Fungal infections were more likely to be identified in pulmonary isolations (23%), while Gram-positive bacteria were more common in abdominal (30%), wound (28%), and blood (38%) isolations (Table 3).

### 3.4. Outcomes Analysis between VAP and Non-VAP Group

Patients in the VAP group required a more extended MV period (*p* = 0.022) and longer ICU stays (*p* < 0.001), and had higher in-hospital (*p* < 0.01) and 30-day mortality rates (*p* < 0.001) compared to the patients with non-VAP (Table 4).

### 3.5. Kaplan–Meier Survival Analysis

In the Kaplan–Meier survival analysis, patients in the VAP group exhibited a significantly lower 30-day survival probability than those in the non-VAP group (*p* < 0.001) (Figure 2).

### 3.6. Multivariable Survival Analysis in the VAP Group

Patients who died within 30 days were more likely to have been diagnosed with VAP five days after starting MV (AOR 3.450, 95% CI 1.200–9.915; *p* = 0.022), received inappropriate antibiotics following a VAP event (AOR 4.083, 95% CI 1.061–15.716; *p* = 0.041), or undergone surgery involving the gastrointestinal system (AOR 4.776, 95% CI 1.287–17.721; *p* = 0.019) (Table 5).

## 4. Discussion

This retrospective study highlighted the incidence and impact of VAP in non-cardiac surgical patients in the ICU. The incidence of VAP in these patients with ICU stays longer than three days was approximately 5.6%. Patients were more prone to develop VAP if they experienced prolonged MV, repeat intubation, lower PaO_2_/FiO_2_ ratios upon ICU admission, or underwent gastrointestinal surgeries. Patients with VAP had significantly longer durations of MV and ICU stays compared to those without VAP. Moreover, the VAP group showed an alarmingly high in-hospital mortality rate of 65%, necessitating increased attention to this condition. In the VAP group, one-half of Gram-negative infections were resistant strains, and one-third were polymicrobial infections. Late-onset VAP, inappropriate antibiotic treatment, and undergoing gastrointestinal system surgery are associated with a higher risk of 30-day mortality.

A mechanical ventilator assists the respiratory system in patients experiencing respiratory failure. Prolonged MV is required when patients need ongoing support due to a lack of readiness for extubation. In 2010, Bouadma et al. implemented a program to prevent VAP in a university tertiary center in Paris. Their study, focusing on primary medical patients, found a higher prevalence of VAP in patients with extended MV during both baseline (22.5% vs. 4.9%) and intervention periods (16.2% vs. 0.8%) [19]. In 2020, Nasreen et al. conducted a retrospective case-control study at a tertiary hospital in Israel focusing on cardiac surgical patients. Patients with longer MV duration were prone to developing VAP (OR 1.138, 95% CI 1.035–1.251; *p* = 0.008) [7]. The bundled care designed to prevent VAP included elements like daily interruption of sedation coupled with spontaneous breathing trials, leading to earlier extubation and reduced VAP occurrences [20]. Consistent with prior literature, this study found that non-cardiac surgical patients requiring MV for more than seven days had a higher risk of developing VAP. Healthcare providers caring for critically ill surgical patients should focus on minimizing the duration of MV and reducing the risk of VAP.

Repeat intubation involves reinserting an endotracheal tube in patients who fail endotracheal extubation. In 1995, Torres et al. conducted a case-control study evaluating the association between repeat intubation and pneumonia in mechanically ventilated patients. Repeated intubation was a substantial risk factor for pneumonia development (OR: 5.94; 95% CI 1.27 to 22.71; *p* = 0.023) [21]. In a propensity-matched study focusing on cardiac surgical populations to assess the cost implications of treating VAP, Luckraz et al. observed that patients requiring repeat intubation had a higher risk of developing VAP than those without [22]. The present study concurs with previous findings, indicating that patients who undergo repeat intubation are at an increased risk of developing VAP. For patients at a higher risk of repeat intubation, healthcare providers may consider alternative strategies, such as opting for tracheostomy before extubation or employing noninvasive ventilation before considering repeat intubation to mitigate the risk of developing VAP.

The PaO_2_/FiO_2_ ratio is a crucial indicator of pulmonary shunt fraction and is widely used to assess the severity of acute respiratory distress syndrome [23]. Sofianou et al. conducted a prospective study in a multidisciplinary ICU in Greece to identify risk factors for acquiring VAP [24]. The researchers found that a PaO_2_/FiO_2_ ratio less than 200 mmHg at the time of ICU admission was significantly associated with the development of VAP. In another retrospective study on a trauma cohort, it was observed that patients were more likely to experience a second episode of VAP if they had a lower PaO_2_/FiO_2_ ratio during the first VAP episode [25]. Similar to previous studies, the current research found that surgical patients with a lower PaO_2_/FiO_2_ ratio upon admission to the ICU were more likely to develop VAP events. Monitoring this ratio can be crucial for the early identification of patients at higher risk of VAP and for implementing targeted preventive measures.

Gastrointestinal surgery involving the gastrointestinal system, liver, biliary tree, and pancreas is considered major abdominal surgery. The literature indicates that surgeries involving the abdomen or pelvis increase the likelihood of postoperative pulmonary complications [26]. The pulmonary complications following abdominal surgery could result from factors like respiratory muscle dysfunction [27] or postoperative aspiration pneumonia [28]. To our knowledge, this is the first study observing that ICU patients undergoing gastrointestinal surgery had an elevated risk of developing VAP. Intensivists and gastrointestinal surgeons should be aware of the VAP risk in this population and implement preventive measures to avoid VAP events.

Moreover, patients undergoing major abdominal surgery and experiencing pulmonary complications tend to have poorer survival outcomes [29]. A systematic review by Sandini et al. highlighted that patients with frailty undergoing such surgeries are at a higher risk of postoperative morbidity and mortality [30]. Consistent with prior research, the study found that patients who had gastrointestinal surgery and developed VAP had a higher mortality risk compared to those without VAP. These findings underscore the dismal outcome in the gastrointestinal surgical patient who develops VAP and call for meticulous treatment in these VAP populations.

Introduced by Mandelli et al. in 1986, the differentiation between early- and late-onset VAP is based on the timing of its development relative to the start of MV [31]. Compared to early-onset VAP, late-onset cases often involve more prolonged periods of MV. This extended support is typically associated with higher disease severity [32,33] and a greater likelihood of acquiring resistant nosocomial infections [34]. In 2002, Moine et al. conducted a multicenter prospective study to evaluate patients with pneumonia and found that those with late-onset pneumonia had a higher mortality risk [35]. Consistent with previous research, the present study discovered that surgical patients with late-onset VAP had a significantly higher mortality rate compared to those with early-onset VAP. These insights highlight the critical importance of timely detection and intervention for VAP, particularly in patients requiring prolonged MV.

A retrospective evaluation of VAP pathogens based on two multicenter clinical trials found that multidrug-resistant bacteria constituted 27.8% in early-onset VAP patients and 32.3% in late-onset VAP patients [36]. Tamayo et al.’s prospective observational study found that polymicrobial infections accounted for 13.7% of VAP patients [6]. In our VAP population, Gram-negative bacteria were the predominant pathogens in pulmonary isolations (73%), followed by fungus (23%) and Gram-positive pathogens (4%). Notably, resistant pathogens (including carbapenem-resistant *Pseudomonas aeruginosa*, carbapenem-resistant *Acinetobacter baumannii*, and *Stenotrophomonas maltophilia*) comprised 48% of the Gram-negative bacteria of VAP pathogens in our study. As critically ill patients in the SICU, VAP patients usually encounter many bacterial infections. Hence, broad-spectrum antibiotics were administered after surgery, which might induce subsequent resistant bacterial pulmonary infection. In addition, polymicrobial infection was observed in 34% of VAP patients. This high prevalence of resistant and polymicrobial infections highlighted the complexity of VAP pathogens.

This study also observed different patterns of bacterial isolation in the abdomen, wound, and blood compared to pulmonary isolations. While Gram-negative bacteria were still predominant (38–49%), Gram-positive bacteria were more common (28–38%) in these sites than in pulmonary isolations (4%). This suggests a diversity of pathogens across different infection sites and potential cross-site pathogen transmission, underscoring the need for precise and effective management strategies tailored to the specific pathogen profile of each patient.

Considering the major association between inappropriate antibiotics and worse outcomes and the different bacteria isolated from different samples, there should be additional emphasis on the importance of cultures. A recent study evaluated this issue and showed the importance of cultures, even in cases where antibiotics were already given [37]. Appropriate antibiotic treatment is crucial in managing patients with infections. Its timely administration is vital to improving patient outcomes [38]. A retrospective study investigating infections caused by multidrug-resistant Gram-negative bacilli post-abdominal surgery found that patients who received appropriate antibiotics had better survival outcomes [39]. Martin-Loeches et al. conducted a prospective observational study to assess ICU-acquired pneumonia’s resistance patterns and outcomes [40]. It was found that patients receiving appropriate antibiotic treatment had significantly higher ICU survival rates (92.9% vs. 82.2%, *p* = 0.03). Consistent with previous research, the current study’s multivariable analysis revealed that patients who received inappropriate antibiotic treatment experienced a higher 30-day mortality rate. The study underscores the necessity of understanding the local microbial environment in healthcare institutions. This knowledge is essential for adequately treating infectious conditions in critically ill surgical patients.

## 5. Limitations

This study faced several limitations. First, being a single-center study, its findings may not be generalizable to other institutions. Second, as a retrospective study, there is a possibility of missing or incomplete confounding factors that were not recorded during the initial data collection. Third, the study did not account for do-not-resuscitate (DNR) orders, which can significantly affect patient outcomes, especially in studies involving critically ill patients. Fourth, the study did not specify the use of traditional or minimally invasive surgical procedures, which may potentially affect postoperative recovery and the risk of VAP. Fifth, because information about qualitative cultures was not collected, there was bias in clarifying VAP and non-VAP patients. The requirement for a positive respiratory sample in the VAP definition could lead to many missing cases. Sixth, the appropriateness of antibiotic therapy initiated empirically was not assessed. Finally, in surgical patients with multiple sites of infection, it is challenging to distinguish between active infection and mere colonization in pulmonary isolates. This can affect the accuracy of diagnosing VAP and understanding its implications. These limitations highlight the need for cautious interpretation of the study’s results and suggest areas for further research.

## 6. Conclusions

Non-cardiac surgical patients who develop VAP face significantly adverse outcomes in the hospital setting. In addition to pulmonary factors, gastrointestinal surgery is associated with VAP development. Surgeons and intensivists need to be acutely aware of the risk factors as well as resistant and polymicrobial infections associated with VAP. The findings stress the importance of timely and appropriate management to enhance survival outcomes for these critically ill surgical patients.

## Figures and Tables

**Figure 1 microorganisms-12-01422-f001:**
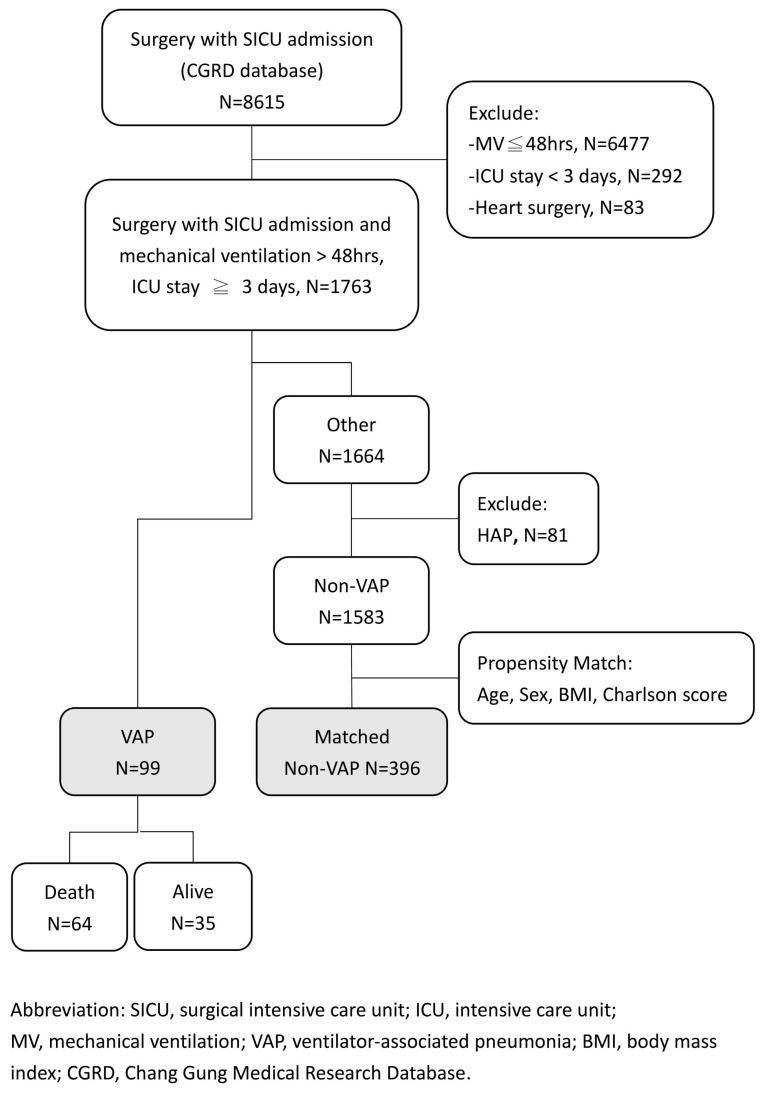
Patient flow chart.

**Figure 2 microorganisms-12-01422-f002:**
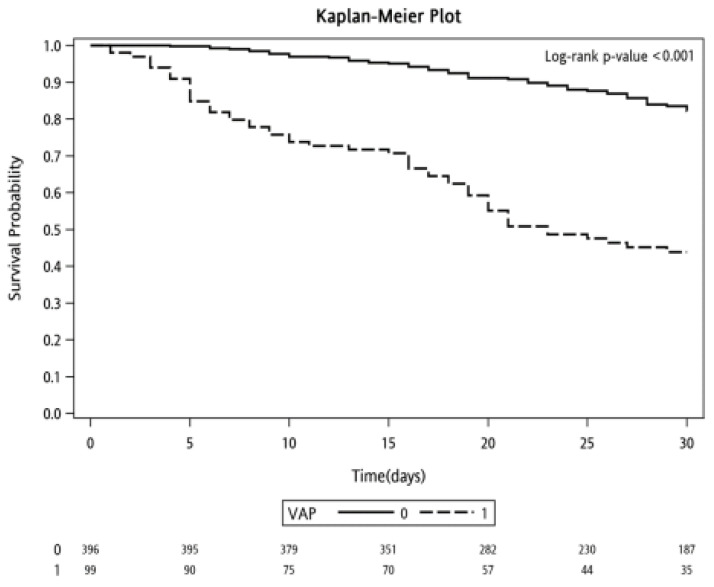
Probability of survival through day 30 in patients with and without ventilator-associated pneumonia.

**Table 1 microorganisms-12-01422-t001:** Demographics, clinical, and laboratory variables between study groups before and after propensity-score matching.

Variables	Pre-Matching Cohort n = 1682			Post-Matching Cohort (1:4) n = 495		
	VAP n = 99	Non-VAP n = 1583	*p*-Value	VAP n = 99	Non-VAP n = 396	*p*-Value
Demographics						
Age, mean (SD) (years)	67.1 (15.1)	66.9 (15.8)	0.893	67.1 (15.1)	66.9 (16.3)	0.909
Age ≥ 65 years, n (%)	62 (60.6)	968 (61.2)	0.914	60 (60.6)	248 (62.6)	0.711
Male, n (%)	73 (73.7)	911 (57.6)	0.002	73 (73.7)	292 (73.7)	1.000
Smoking, n (%)	27 (27.3)	397 (25.1)	0.626	27 (27.3)	121 (30.6)	0.523
BMI, mean (SD) (kg/m^2^)	24.1 ± 5.6	24.0 ± 6.0	0.953	24.1 ± 5.6	24.0 ± 5.1	0.600
Patient referred from ER, n (%)	19 (19.2)	389 (24.6)	0.226	19 (19.2)	102 (25.8)	0.174
Pre-ICU stay, mean (SD) (days)	6.2 (8.6)	7.0 (15.7)	0.356	6.2 (8.6)	6.8 (16.2)	0.568
SOFA score, mean (SD) *	6.8 (3.5)	6.0 (3.7)	0.026	6.8 (3.5)	6.0 (3.6)	0.039
Comorbidities						
Charlson score, mean (SD)	2.0 (2.1)	2.1 (2.0)	0.545	2.0 (2.1)	2.0 (2.0)	0.892
Charlson score ≥ 3, n (%)	34 (34.3)	586 (37.0)	0.593	34 (34.3)	136 (34.3)	1.000
Myocardial infarction, n (%)	3 (3.0)	66 (4.2)	0.786	3 (3.0)	15 (3.8)	0.734
Congestive heart failure, n (%)	5 (5.1)	171 (10.8)	0.07	5 (5.1)	25 (6.3)	0.638
Cerebrovascular disease, n (%)	11 (11.1)	253 (16.0)	0.197	11 (11.1)	67 (16.9)	0.156
Chronic pulmonary disease, n (%)	23 (23.2)	248 (15.7)	0.047	23 (23.2)	66 (16.7)	0.128
Diabetes Mellitus, n (%)	21 (21.2)	407 (25.7)	0.319	21 (21.2)	96 (24.2)	0.526
Liver disease, n (%)	20 (20.2)	313 (19.8)	0.917	20 (20.2)	74 (18.7)	0.731
Cancer, n (%)	34 (34.3)	569 (35.9)	0.747	34 (34.3)	133 (33.6)	0.887
Interventions						
MV > 7 days, n (%)	61 (61.6)	438 (27.7)	<0.001	61 (61.6)	112 (28.3)	<0.001
Tracheostomy, n (%)	6 (6.1)	75 (4.7)	0.341	6 (6.1)	23 (5.8)	0.924
Repeat intubation < 14 days, n (%)	28 (28.3)	247 (15.6)	<0.001	28 (28.3)	68 (17.2)	0.012
Hemodialysis, n (%)	15 (15.2)	257 (16.2)	0.776	15 (15.2)	64 (16.2)	0.806
ECMO, n (%)	4 (4.0)	20 (1.3)	0.048	4 (4.0)	7 (1.8)	0.242
Central venous catheter, n (%)	77 (77.8)	1324 (83.6)	0.129	77 (77.8)	330 (83.3)	0.196
Total parenteral nutrition, n (%)	43 (43.4)	501 (31.7)	0.015	43 (43.4)	120 (30.3)	0.013
Double lumen, n (%)	9 (9.1)	150 (9.5)	0.899	9 (9.1)	37 (9.3)	0.938
Urinary catheter, n (%)	93 (93.9)	1512 (95.5)	0.467	93 (93.9)	377 (95.2)	0.608
Drainage catheter, n (%)	87 (87.9)	1385 (87.5)	0.910	87 (87.9)	337 (85.1)	0.481
Blood transfusion, n (%)	87 (87.9)	1426 (90.1)	0.479	87 (87.9)	360 (90.9)	0.362
ICU readmission, n (%)	20 (20.2)	280 (17.7)	0.526	20 (20.2)	65 (16.4)	0.371
Pre-exposed Rx						
Antibiotics ^a^ (≥2 days)	68 (68.7)	973 (61.5)	0.151	68 (68.7)	256 (64.7)	0.450
Carbapenem ^a^, n (%)	38 (38.4)	634 (40.1)	0.743	38 (38.4)	170 (42.9)	0.413
APP or APC ^a^, n (%)	36 (36.4)	568 (35.9)	0.923	36 (36.4)	145 (36.6)	0.963
Fluoroquinolone ^a^, n (%)	17 (17.2)	264 (16.7)	0.898	17 (17.2)	78 (19.7)	0.568
Chemotherapy ^b^, n (%)	7 (7.1)	94 (5.9)	0.645	7 (7.1)	21 (5.3)	0.496
IMS ^b^ or Steroid ^b,c^, n (%)	17 (17.2)	151 (9.5)	0.014	17 (17.2)	4 (10.6)	0.071
Surgical details						
Critical condition, n (%) **	59 (59.6)	744 (47.0)	0.015	59 (59.6)	183 (46.2)	0.017
Wound class ≥ 3, n(%)	60 (60.6)	929 (58.7)	0.707	60 (60.6)	227 (57.3)	0.554
ASA ≥ 3, n (%)	91 (91.9)	1370 (86.5)	0.127	91 (91.9)	345 (87.1)	0.188
ASA ≥ 4, n (%)	28 (28.3)	375 (23.7)	0.299	28 (28.3)	102 (25.8)	0.610
Op duration, mean (SD) minutes	281.6(171.7)	281.9 (192.1)	0.987	281.6 (171.7)	272.7(177.0)	0.653
Repeat Op before VAP, n (%)	36 (36.4)	535 (33.8)	0.601	36 (36.4)	153 (38.6)	0.677
Surgical site						
-Gastrointestinal surgery, n (%)	77 (77.8)	1097 (69.3)	0.075	77 (77.8)	267 (67.4)	0.045
-Gastrointestinal tract, n (%)	68 (68.7)	956 (60.4)	0.101	68 (68.7)	234 (59.1)	0.080
-HPB, n (%)	22 (22.2)	257 (16.2)	0.120	22 (22.2)	57 (14.4)	0.057
-Abdominal wall, n (%)	15 (15.2)	251 (15.9)	0.852	15 (15.2)	60 (15.2)	1.000
-Urology (including kidney), n (%)	8 (8.1)	227 (14.3)	0.081	8 (8.1)	53 (13.4)	0.151
-Chest, n (%)	3 (3.0)	83 (5.2)	0.332	3 (3.0)	19 (4.8)	0.591
-Limbs, n (%)	13 (13.1)	246 (15.5)	0.519	13 (13.1)	66 (16.7)	0.390
-Others, n (%)	1 (1.0)	9 (0.6)	0.458	1 (1.0)	3 (0.8)	1.000
Lab data						
PaO_2_/FiO_2_ mmHg, mean (SD) *	280.3(116.6)	338.5 (130.6)	<0.001	280.3 (116.6)	348.6 (153.6)	<0.001
Albumin < 3.5 mg/dL, n (%) *	64 (64.7)	958 (60.5)	0.414	64 (64.7)	244 (61.6)	0.578
ALT > 80 U/L, n (%) *	12 (12.1)	149 (9.4)	0.374	12 (12.1)	43 (10.9)	0.721
Creatinine > 1.2 mg/dL, n (%) *	47 (47.5)	649 (41.0)	0.204	47 (47.5)	179 (45.2)	0.685
Hemoglobin < 8 g/dL, n (%) *	8 (8.1)	97 (6.1)	0.436	8 (8.1)	28 (7.1)	0.729
Platelet < 150 1000/μL, n (%) *	32 (32.3)	408 (25.8)	0.150	32 (32.3)	110 (27.8)	0.371
Total bilirubin > 1.4 mg/dL, n (%) *	26 (26.3)	322 (20.3)	0.158	26 (26.3)	87 (22.0)	0.363

VAP, ventilator-associated pneumonia; SD, standard deviation; BMI, body mass index; ER, emergency room; ICU, intensive care unit; SOFA, Sequential Organ Failure Assessment; MV, mechanical ventilation; ECMO, extracorporeal membrane oxygenation; Rx, treatment and medications; APP, antipseudomonal penicillin; APC, antipseudomonal cephalosporin; IMS; immunosuppressive agents; ASA, American Society of Anesthesiologists; HPB, Hepato-biliary-pancrease; PaO_2_/FiO_2_, partial pressure of oxygen in arterial blood/fraction of inspired oxygen; ALT, *Alanine transaminase*. * Upon ICU admission. ** Referred to unstable vital signs during operation. ^a^ Exposed within last 14 day. ^b^ Used within the last 30 days. ^c^ Dose ≥ 10 mg/day prednisolone-equivalent and ≥7 days.

**Table 2 microorganisms-12-01422-t002:** Multivariable analysis for ventilator-associated pneumonia in surgical intensive care unit.

Variable	Adjusted Odds Ratio	95% CI	*p* Value
Total parenteral nutrition	0.918	0.518–1.627	0.770
Critical condition *	1.453	0.775–2.721	0.244
MV > 7 days	6.435	3.15–13.146	<0.001
Repeat intubation < 14 days	6.438	2.934–14.127	<0.001
Gastrointestinal surgery	2.257	1.13–4.507	0.021
SOFA Score **	0.972	0.887–1.064	0.534
PaO_2_/FiO_2_ (every increased 10 mmHg) **	0.950	0.992–0.998	<0.001

CI = confidence interval; MV = mechanical ventilation; SOFA = Sequential Organ Failure Assessment; PaO_2_/FiO_2_: partial pressure of oxygen in arterial blood [PaO_2_]/fraction of inspired oxygen. * Referred to unstable vital signs during operation. ** Upon ICU admission.

**Table 3 microorganisms-12-01422-t003:** Distribution of bacterial pathogens in surgical patients with VAP.

Bacterial Pathogen	Pulmonary Isolation n = 175	Abdominal Isolation n = 150	Wound Isolation n = 62	Blood Isolation n = 68
Gram-positive bacteria	7 (4%)	54 (30%)	20 (28%)	28 (38%)
*Staphylococcus or Streptococcus*				
*Staphylococcus aureus*	4	1	0	4
*Staphylococcus* sp.	1	18	8	18
*Streptococcus pneumoniae*	1	0	0	0
*Streptococcus* sp.	1	4	2	1
*Enterococcus*				
*Enterococcus faecalis*	0	12	5	1
*Enterococcus faecium*	0	10	2	1
*Enterococcus* sp.	0	3	2	0
*Other Gram-positive bacteria*	0	6	1	3
Gram-negative bacteria	127 (73%)	89 (49%)	33 (46%)	28 (38%)
*Enterobacteriaceae*				
*Escherichia coli*	1	16	6	6
*Klebsiella pneumonia*	7	7	4	5
*Proteus mirabilis*	0	4	1	0
*Enterobacter species*	4	8	5	2
*NFGNB*				
*Pseudomonas aeruginosa*	31	14	4	4
*CRPA*	16	9	1	1
*Acinetobacter baumannii*	16	5	1	2
*CRAB*	12	4	1	2
*SM*	33	11	3	1
*Other GNB*	7	11	7	5
Anaerobic	0	15 (8%)	8 (11%)	5 (7%)
NTM	1 (1%)	0	0	0
Fugus (*candida* sp., *mold*, *yeast*)	40 (23%)	23 (13%)	11 (15%)	12 (16%)
Polymicrobial (*include yeast*)	34/99 (34%)	42/99(42%)	20/99(20%)	16/99(16%)

VAP, ventilator-associated pneumonia; NFGNB, non-fermenting Gram-negative bacteria; CRPA, carbapenem-resistant *Pseudomonas aeruginosa*; CRAB, carbapenem-resistant *Acinetobacter baumannii*; SM, *Stenotrophomonas maltophilia*; NTM, nontuberculous mycobacteria.

**Table 4 microorganisms-12-01422-t004:** Comparison of outcomes of VAP and non-VAP patients.

Hospital Outcomes	VAP N = 99	Non-VAP N = 396	*p* Value
MV days, mean (SD) (days)	16.9 (15.3)	12.7 (16.5)	0.022
ICU stay, mean (SD) (days)	17.1 (14.3)	15.3 (14.6)	<0.001
Hospital stay, mean (SD) (days)	31.4 (29.7)	37.4 (31.3)	0.084
In-hospital mortality, n (%)	64 (64.7)	101 (25.5)	<0.001
30-day mortality, n (%)	54 (54.6)	55 (13.9)	<0.001

VAP, ventilator-associated pneumonia; MV, mechanical ventilator; SD, standard deviation; ICU, intensive care unit.

**Table 5 microorganisms-12-01422-t005:** Multivariable analysis for 30-day mortality in VAP patients.

Variable	Adjusted Odds Ratio	95% CI	*p* Value
Age, years	1.022	0.987–1.059	0.217
Pre-ICU stay, days	1.050	0.963–1.146	0.267
Male	0.400	0.110–1.454	0.164
Immunosuppressant ^a^ or Steroid ^a,b^	7.586	0.753–76.468	0.086
Cancer	2.444	0.692–8.629	0.165
Late-onset VAP ^c^	3.450	1.200–9.915	0.022
Inappropriate antibiotics treatment	4.083	1.061–15.716	0.041
Gastrointestinal surgery ^d^	4.776	1.287–17.721	0.019

VAP, ventilator-associated pneumonia; CI, confidence interval; ICU, intensive care unit. ^a^ Used within the last 30 days. ^b^ Dose >= 10 mg/day prednisolone-equivalent and >= 7 days. ^c^ Late-onset VAP: VAP onset five days after mechanical ventilation. ^d^ Gastrointestinal surgery includes gastrointestinal and hepato-.

## Data Availability

The original contributions presented in the study are included in the article, further inquiries can be directed to the corresponding author.

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
