# Peer review of "Risk Factors, Pathogens, and Outcomes of Ventilator-Associated Pneumonia in Non-Cardiac Surgical Patients: A Retrospective Analysis"

_microorganisms, 2024, doi:10.3390/microorganisms12071422_

Round 1
Reviewer 1 Report
Comments and Suggestions for Authors
Thank you for the opportunity to review this manuscript. The authors evaluate the predictors and outcomes of intubated patients with VAP after non-cardiac surgery. The authors used a propensity score matching and also assessed the culture results and related treatments. The article addresses an important topic, is well-written and performed, and emphasize specific issues for clinical practice. Overall, the methods and analyses performed are excellent. Some minor issues that should be addressed by the authors to further improve the manuscript:
1. Methods – the data is taken from a retrospective registry. Yet, the definition of VAP is thorough and include different factors which are hard to assess without entering patients' medical records. Please explain how the diagnosis was made – by the research team or based on ICD codes or else?
2. The fact that a positive respiratory sample is needed for the definition of VAP could lead to many missing cases, which should be stated in the limitations.
3. Statistical analysis – more data on the matching process is needed.
4. Discussion – considering the major association between inappropriate antibiotics and worse outcomes and the different bacteria isolated from different samples, there should be additional emphasis on the importance of cultures. A recent study evaluated this issue and showed the importance of cultures, even if antibiotics were already given (PMID: 37884696). I suggest addressing this issue, possibly with the example above.
Comments on the Quality of English Language
None specific
Author Response
please see th attachment

Reviewer 2 Report
Comments and Suggestions for Authors
This is a god study evaluating the outcomes of VAP patients and the risk factors associated with mortality. Several comments need to be addressed as follows:
1. Line 23: Add the abbreviation "ICU" after "intensive care unit" since you used the abbreviation on line 26.
2. Lines 17, 29, and throughout the manuscript: I suggest replacing "digestive" with "gastrointestinal" as the latter is more common clinically, unless the authors think otherwise.
3. Line 73: You don't need to spell out ICU as you've already spelled it out in the introduction. Same for VAP on line 75 and 241. Also, same for MV on line 244.
4. Section 2.3. Definitions: It's better to list the terms in bullet points to make it more organized.
5. Line 135: The ethics statement is better placed under a separate subsection or at the end of the 2.1 section.
6. Line 146: What was the P value of the univariate analysis required to include the variables in the multivariable logistic regression? In other words, based on what did you select the variables you included in the multivariable regression?
7. Line 149: Did you also utilize log-rank test to generate the P value.
8. Results: Please number the subsections under this manjor section.
9. Results (line 153): During which period was this number of patients admitted?
10. Figure 1: Replace "Exclude" with "Excluded." Also, list the abbreviations next to the figure legend.
11. Table 1: Correct "bilirubine" to "bilirtubin"
12. Lines 182, 190, and 228: Replace "Multivariate" with "Multivariable" as this kind of regression analysis is considered multivariable and not multivariate. This is a common misunderstanding among researchers. You may refer to this article to understand the difference between the two types of regression analyses: https://www.ncbi.nlm.nih.gov/pmc/articles/PMC3518362/
13. Under "Distribution of Bacterial Pathogens According to Infection Sites in the VAP Group", please italicize the names of all bacterial pathogens mentioned.
14. Lines 229-232: Replace "OR" with "AOR" or "aOR" however you would liket o report adjusted odds ratio. That's because plain OR usually refers to that resulting from univariate regression analysis.
15. Line 228: What about assessing the causative pathogen, the resistance profile (MDR, XDR, DTR, etc), or the antibiotic received as potential factors associated with mortality? If it is not possible to assess such correlation, please add this to your limitations.
16. Limitations (line 354): Add to the limitations that the appropriateness of antibiotic therapy initiated empirically was not assessed. You also did not assess whether targeted (definitive) therapy was appropriate based on culture and susceptibility results. Both of these are important factors that could have impacted the outcomes.
Round 2
Reviewer 1 Report
Comments and Suggestions for Authors
Thank you for the opportunity to review this paper once again. The authors have answered all my comments and the manuscript has significantly improved. Thank you and good luck.
Comments on the Quality of English Language
None specific.
Author Response
please see the atchment

Reviewer 2 Report
Comments and Suggestions for Authors
Thank you to the authors for addressing he comments and improving their manuscript. However, some comments were still not addressed. Please revise these comments and make sure you address them.
1. Comment #6: You still didn't mention the P value as you claimed in your response.
2. Comment #7 (mentioning log-rank test under the statistical analysis section).
3. Comment #9: I still can't see the patient admission period. You said it's on line 153, but this line corresponds to Figure 1. It should be on lines 148-149.
4. Comment #16: I couldn't find that you added the suggested limitations to the limitations paragraph (the appropriateness of antibiotic therapy initiated empirically).
